# New Insight into Selective Serotonin Receptor Agonists in the Central Nervous System, Studied with WAY163909 in Obese and Diabetic Wistar Rats

**DOI:** 10.3390/brainsci13040545

**Published:** 2023-03-25

**Authors:** Ivaylo Bogomilov, Nadka Boyadjieva, Rumen Nikolov

**Affiliations:** Department of Pharmacology and Toxicology, Medical University of Sofia, st Zdrave 2, 1000 Sofia, Bulgaria

**Keywords:** obesity, diabetes, insulin resistance, serotonin, central nervous system, agonist

## Abstract

Background and aims: We investigated the effect of WAY-163909, a novel 5-hydroxytryptamine selective 2C receptor agonist on body weight, blood glucose levels, and insulin resistance in obese and diabetic Wistar rats. Materials and methods: We used twenty male Wistar rats with obesity and obesity-induced diabetes and twenty healthy Wistar rats as a control group. Each of these groups was separated into two subgroups: one with a daily intraperitoneal application of WAY-163909 (1 mg/kg) and one without. During the study, body weight, blood glucose levels, and immunoreactive insulin were tracked. Results: A reduction of 5.5% (*p* < 0.05) in body weight was registered in the rat group with diabetes and obesity and 2.56% in the control group with a daily application of WAY-163909 (1 mg/kg) at the end of the study. Decreases of 35.4% in blood glucose levels at week four in the diabetic and obese rat group with a daily application of WAY-163909 (1 mg/kg) were registered. A reduction of insulin levels of 4.1% (*p* < 0.05) in the diabetic and obese rats group using WAY-163909 was also observed. Conclusion: In our study, using WAY-163909 (1 mg/kg) led to a reduction of blood glucose levels, immunoreactive insulin, and body weight.

## 1. Introduction

Obesity can lead to various metabolic diseases, such as type 2 diabetes, dyslipidemia and hyperuricemia, which are serious medical problems, especially in developed countries [1]. In the majority of cases, obesity is developed by increased caloric intake and decreased energy expenditure. Over the last few years, there has been an increase in obesity worldwide, which is related to the changing eating habits in modern societies [2]. Foods rich in both fats and sugars are part of most people’s diets. Such foods were among the main risk factors for the development of obesity and insulin resistance [3]. Interestingly, despite the increase in the health culture of the population, especially in developed countries, obesity is becoming a bigger problem [4,5,6,7,8]. From this, it can be concluded that food intake is most likely not only a volitional process; different neurobiological interactions have a role in the excessive intake of nutrients. High-fat, high-carbohydrate diet in rodents leads to an increased concentration of neuropeptide Y and a decrease in concentrations of proopiomelanocortin in the arcuate nucleus in the central nervous system (CNS), which would lead to hyperphagia [9]. It can be concluded that the intake of certain types of foods directly interact with gene expression in the hypothalamus. A trend in modern societies is to increase the number of snacks with various types of foods rich in fats and carbohydrates, such as chips, sweets, and sweetened drinks [10,11]. This type of intermediate meals, which do not ensure a full nutritional regime and do not exclude the basic meals during the day, is the basis of taking additional calories and a reason for the development of obesity [10,11]. Such changes have also been observed in rats that are given the opportunity to eat unlimited foods rich in fats and sugars. These animals do not increase meal volume but increase the number of meals, which in turn leads to obesity and insulin resistance [12,13].

Eating is a complex neurobiological process involving multiple organs and systems. In order to control food intake and to be able to respond to the body’s energy needs, various signals from almost all organs and systems are integrated into the brain. This flow of information must be processed and systematised to make good decisions and ensure the normal course of all homeostatic processes. The integration of information related to nutrition into the central nervous system (CNS) is controlled primarily by the hypothalamus. Many different nuclei in the hypothalamus and mesocorticolimbic system contribute to the proper regulation of food intake and eating behaviours. The blood-brain barrier surrounding the hypothalamic arcuate nucleus is thinned, and from there, different hormones from the periphery can pass and regulate the necessary processes related to nutrition [14,15]. The arcuate nucleus receives projections from many other areas, and neuropeptide Y (NPY)/agouti (AgRP) is primarily responsible for stimulating food intake. The peptides that stimulate food intake are called orexigenic. In turn, anorexigenic peptides, such as proopiomelanocortin (POMC), are involved in ending the act of eating. The paraventricular nucleus is another area in the hypothalamus that has an important role in the integration of nutritional needs and energy balance [16]. The paraventricular nucleus receives signals from numerous other nuclei in the CNS and from the periphery.

Serotonin (5-HT, 5-hydroxytryptamine) is one of the neurotransmitters in the CNS, which has an important role in the hypothalamus and in the mesocorticolimbic system to regulate food intake and energy balance. Serotonin is formed from the amino acid tryptophan in the nuclei of the brain. Serotonergic fibres give projections to multiple areas in the central nervous system, thereby integrating information from almost all units responsible for food intake, including the striatum, amygdala, thalamus, hypothalamus, and mesocorticolimbic area [17]. The extracellular concentration of serotonin is controlled by the serotonin transporter (SERT), located on the presynaptic neuronal surface [18]. SERT is an important element in the overall serotonin mediation. Studies have established its ubiquitous role in the processes of controlling eating behaviours, and because SERT can be visualised with various methods, its level in the body can be traced [18].

Multiple neuronal pathways are affected by obesity. However, the main pathways implicated in obesity are those related to dopamine and serotonin secretion and mediation. The first studies of central brain serotonin mediation showed that lower levels in the hypothalamus lead to hyperphagia and obesity in experimental animals [19,20,21,22]. In turn, the infusion of serotonin into the hypothalamus can lead to a reduction in body weight in rodents [22]. In experimental models in rodents and in humans, damaging SERT leads to obesity, which is associated with increased waist circumference and an accumulation of visceral adipose tissue and insulin resistance [23,24,25]. FDA-approved drugs, such as a selective 5-HT2C receptor agonist called lorcaserin, were used in humans for the treatment of obesity [26]. In addition, fenfluramine is a drug that has also been used to treat metabolic syndrome and obesity [24]. Experience with selective serotonin reuptake inhibitors (SSRIs) also suggests that there may be a change in body weight in people that use them [27,28,29]. Some of the available SSRIs lead to an increase in body weight, while others lead to a reduction. Overall, the net effect is associated with an increase in body weight [28,29,30]. Furthermore, different genetic studies in humans show that carriers of a short allele for the SERT gene are at risk for the development of metabolic syndrome, and one of its hallmarks is the visceral type of obesity [31]. Some studies show a clear correlation between body weight and SERT concentration in the mesocorticolimbic system [32].

The relationship between serotonin mediation in the central nervous system and carbohydrate metabolism is demonstrated in some research. In rodents, extracellular serotonin levels in the suprachiasmatic nucleus have been found to be associated with changes in blood glucose levels and the development of insulin resistance [33]. In addition, agonising serotonin 5-HT2C receptors can lead to an improvement in glucose metabolism [34], and its antagonism can lead to the development of type 2 diabetes and insulin resistance [35]. In rodent models with a defect in SERT, hyperphagia and the development of obesity and even type 2 diabetes with marked hyperinsulinemia were observed [19]. In human genetic studies, it was found that serotonin mediation in CNS is responsible for the development of metabolic syndrome, insulin resistance, and type 2 diabetes, as it is associated with polymorphism in SERT genes [36]. It is also known that long-term use of SSRIs leads to body weight gain provoked by excessive food intake and the development of glucose intolerance and even type 2 diabetes [37]. However, until now, there was not a compiled study that showed the effect of serotonin mediation in the context of obesity and obesity-induced diabetes. In our study, we use WAY-163909-a, selective serotonin 2C receptor agonist, in the central nervous system to investigate its effect on body weight, blood sugar levels, and insulin resistance in obese and diabetic Wistar rats.

## 2. Materials and Methods

### 2.1. Experimental Animals

We used sexually mature male Wistar rats for the purposes and tasks of the experimental studies. Experiments were conducted under standard rearing and food and water conditions. The experiments were performed according to the International Guiding Principles for Animal Research, and the ethical principles in planning and conducting the experiments were according to the Commission on Ethics of Scientific Research at the Medical University of Sofia.

### 2.2. Developing an Experimental Model Investigating the Role of Central Brain Serotonin in Obese and Diabetic Wistar Rats Using the Selective Serotonin Receptor Agonist, WAY163909

The in vivo experiments were carried out on forty male Wistar rats with an average weight of 220–240 g at the time animals were taken from a central vivarium. During the experiments, the circadian rhythm (12 h. light/12 h. dark) was observed. The average temperature maintained during the experiments was 22 C. The rats were divided into two groups: rats with induced obesity and diabetes and healthy rats (control group).

Induced obesity was achieved by administering a high-calorie diet that included high-fat, high-carbohydrate foods (cafeteria diet) and high fructose corn syrup for 8 weeks, compared with a standard diet regimen for the control group. During the period aimed to induce obesity, the experimental animals were given high-fat, high-carbohydrate food. Experimental animals in the obesity group had access to food throughout the day, and their food intake was only restricted in the hours before taking blood samples. In addition to a diet high in fat and carbohydrates, mainly conditioned by various types of nuts and foods with refined sugars and fats (chips and sweets), experimental animals were given a glucose-fructose syrup, high fructose corn syrup, which, according to other authors, leads to the rapid development of obesity and type 2 diabetes within 2 to 10 weeks in experimental animals, including Wistar rats [36].

The cafeteria diet included high-fat and high-carbohydrate food (fat 30%, carbohydrate 57%, and protein 13% in total; the energy density of the combined high-lipid, high-carbohydrate food was 4298 kcal/kg; the control group was given 2908 kcal/kg) and the glucose-fructose syrup, high fructose corn syrup (HFCS; 55-55% fructose, 42% glucose, and 3% higher saccharides). In Table 1 can be found the composition and nutritional value of used food in the cafeteria diet.

In the obese group, all animals developed fasting blood glucose levels (after at least 8 h of fasting) above 7.0 mmol/L, which is defined as impaired fasting glycemia, and 19 of them above 7.5 mmol/L, which is defined as diabetes mellitus [38].

All rats in the obese group reached a BMI (body mass index) above 0.68 g/cm^2^, which is accepted for obesity in Wistar adult male rats [39]. All rats in the control group had a BMI between 0.48 g/cm^2^ to 0.68 g/cm^2^, which is defined as the normal value for adult Wistar rats [39].

BMI was calculated according to the formula BMI = weight in grams/length^2^ in cm.

Each of these two groups was divided into two: one with a daily intraperitoneal application of WAY163909 (1 mg/kg) within 4 weeks, and one without (Figure 1). The substance, WAY-163909, acts as a selective agonist for the serotonin 5-HT2C receptor.

The animals were observed for 4 weeks. The animals were divided into groups of three or four per cage. The body mass of all animals was measured weekly, along with blood glucose and immunoreactive insulin levels.

### 2.3. Biochemical Methods

#### 2.3.1. Determination of Glucose Levels in the Blood of Rats

Blood glucose levels were determined from a cut to the tip of the tail of the rat and gently massaging the tail from the base upwards to generate a blood droplet. This droplet was captured on a glucose strip placed inside a handheld glucometer (FreeStyle Optium Neo, Abbott Diabetes Care Ltd., Maidenhead, Berkshir, UK), which normally requires 0.5–2 µL of blood, and the result was obtained after 3 s on the glucometer screen in mmol/L.

#### 2.3.2. Determination of Insulin Resistance by Calculating the HOMA-Index

To calculate the insulin sensitivity in the experimental animals, the HOMA-index (Homeostatic Model for the Assessment of Insulin Sensitivity) was used. Assessment data are based on results from physiological studies that describe the feedback loop of glucose regulation and insulin secretion. The mathematical formula created in this way allows us to calculate the function of the beta cells and determine insulin resistance.

The index was calculated according to the following formula: HOMA = (Glucose levels × Insulin levels)/22.5

From each rat and in all groups, blood was collected following the rules for handling experimental animals. Blood glucose levels were determined from a cut to the tip of the tail of the rat and gently massaging the tail from the base upwards to generate a blood droplet. (FreeStyle Optium Neo, Abbott Diabetes Care Ltd., Maidenhead, Berkshire UK). This droplet was captured on a glucose strip placed inside a handheld glucometer (Free-Style Optium Neo, Abbott Diabetes Care Ltd., Maidenhead, Berkshire, UK), which normally requires 0.5–2 µL of blood. The result was obtained after 3 s on the glucometer screen in mmol/L. To determine insulin in the blood of animals, RAT INSULIN ELISA (BioVendor, Brno, Czech Republic) was used (the methodology of the test is described below).

### 2.4. ELISA (Enzyme Immunosorbent Assay) Method

Sampling blood for the measurement of rat insulin levels was done from the lateral tail vein of the rat, following the Lee G and Goosens KA protocol. Blood samples were collected from the trunk in EDTA tubes for the measurement of insulin levels. All blood samples were centrifuged (2500 rpm, 20 min, LC-04B).

The technology uses two high-affinity monoclonal antibodies in an immunometric assay system. This assay is based on a two-step procedure. In the first step, the standards and samples are incubated in streptavidin-coated wells with a biotin-labelled monoclonal antibody (capture antibody). During a 1 h incubation period with continuous agitation, the captured antibody-antigen complex is developed and immobilised on the reactive surface of wells. Afterwards, incubation wells are washed repeatedly. In the second step, the horseradish-peroxidase-(HRP)-labelled monoclonal antibody (signal antibody) is added. It binds to an epitope of the insulin molecule different from that recognised by the capture antibody, developing the formation of a capture-antibody-antigen-signal antibody complex, also referred to as a ‘sandwich’. After the 1 h incubation period with continuous agitation, the reaction mixture is washed repeatedly. After the addition of a ready-to-use tetramethylbenzidine (TMB) peroxide substrate, the signal is measured in an ELISA photometer at 450 nm and 405 nm wavelengths. The concentration of antigens is directly proportional to the optical density measured in the wells. The unknown concentration of rat insulin in samples is read off a calibration curve constructed by plotting binding values against a series of calibrators containing a known amount of rat insulin. 

### 2.5. Statistical Method

Data from all experiments were analysed with SPSS software, v. 28. Data were presented as means ± standard deviation. Differences between groups were analysed by one-way analysis of variance (ANOVA). Differences were considered statistically significant at *p* < 0.05. The obtained results were presented graphically using the statistical program SPSS 28.0.0.0 and Microsoft Excel 2003. 

## 3. Results

### 3.1. Effect on Body Weight in Obese and Diabetic Wistar Rats Using a Selective Serotonin Receptor Agonist, WAY163909

In the study, the body weight values of the experimental animals were monitored every week in all groups. From the measurements, a significant decrease in body weight was found in the groups with daily intraperitoneal administration of WAY163909 (1 mg/kg). Group 1 contained rats with obesity and type 2 diabetes and the daily intraperitoneal application of WAY163909 (1 mg/kg); a significant reduction in body weight was observed at the end of the four-week period. With a *p* < 0.05, rats moved from an initial weight of 435 ± 20.4 g to 411 ± 12.6 g at the end of week four (Table 2 and Figure 2). There was also a statistically significant difference when comparing the values of body weight between group 1 and group 2 at the end of the study, *p* < 0.01 (Figure 2). In addition, a reduction in body weight was also observed in the control group with the intraperitoneal application of WAY163909 (1 mg/kg); group 3 moved from an initial weight of 340 ± 24.9 g to 331 ± 7.33 g at the end of week four, which was not a statistically significant change. Comparing the two control groups, 3 and 4, at the end of the fourth week, a statistically significant difference in body weight values was found between the group using WAY163909 (1 mg/kg) and the group without daily application (*p* < 0.05) (Figure 2). There were no statistically significant differences between these groups at the beginning of the study, *p* = 0.56. 

### 3.2. Effect on Glucose Levels in Obese and Diabetic Wistar Rats Using a Selective Serotonin Receptor Agonist, WAY163909

From the conducted studies, a decrease in the fasting blood glucose levels was found in group 1, which included rats with obesity and type 2 diabetes with the daily intraperitoneal application of WAY163909 (1 mg/kg). The average blood glucose levels at the beginning of the study in group 1 were 11.6 ± 2.24 mmol/L, and the reduction in levels of blood glucose at the end of the four-week follow-up dropped to 7.5 ± 0.92, which is a statistically significant decrease, *p* < 0.05 (Table 3, Figure 3). There were no significant differences in blood glucose levels between group 1 and group 2 at the beginning of the study. In the fourth week of the study, there was a statistically significant difference in blood glucose levels between group 1 and group 2, which contained the groups with obese and diabetic rats, *p* < 0.01 (Figure 3). No statistically significant decrease in blood glucose levels was observed in the other three groups. During the study period, no hypoglycemic levels were recorded in any of the groups (Table 3 and Figure 3).

### 3.3. Effect on the Levels of Immunoreactive Insulin and Insulin Resistance, Calculated by the HOMA-Index, in Obese and Diabetic Wistar Rats Using a Selective Serotonin Receptor Agonist, WAY163909

Levels of immunoreactive insulin in the different groups were monitored weekly. From the measurements carried out, a significant decrease in immunoreactive insulin was found in group 1, the group with obesity and type 2 diabetes, with the daily intraperitoneal application of WAY163909 (1 mg/kg), *p* < 0.01. They moved from 18.3 ± 4.1 mIU/mL initially to 10.6 ± 1.18 mIU/mL at the end of week four (Table 4 and Figure 4). In addition, a statistically significant difference was also found when comparing the values of immunoreactive insulin in groups 1 and 2, *p* < 0.01, at the end of the fourth week (Figure 4). No significant dynamics were observed in the level of immunoreactive insulin in the other groups (Figure 4). The same constellation was observed in the follow-up of the HOMA-index, with a statistically significant difference in HOMA-index values between groups 1 and 2 at the end of the 4-week observation period. This speaks of a decrease in insulin resistance values in the group using the selective serotonin receptor agonist (Table 5 and Figure 5). This was due to the decreased blood glucose levels described above and the decreased immunoreactive insulin levels during the follow-up. Thus, HOMA-index values in group 1 moved from 9.43 ± 1.686 at baseline to 3.53 ± 0.37 at the end of the fourth week, which was comparable with the control groups’ levels (Figure 5).

## 4. Discussion

Obesity and the metabolic disorders associated with it have become a pandemic in recent decades [40]. Many world medical organisations recognise obesity as a chronic relapsing disease. For this reason, obesity, and the metabolic disorders it causes, are objects of scientific interest worldwide [41]. Finding the right approaches in the overall process of diagnosis and treatment of metabolic diseases and obesity is becoming increasingly important in modern life. One of the main risk factors for developing type 2 diabetes is obesity. The increased cardiovascular risk caused by obesity and the diseases associated with it, such as disorders in glucose metabolism, disorders in lipid metabolism, and high blood pressure, are the immediate causes of disability and mortality, not only among the elderly population but among adolescents in developed countries.

Central brain serotonin mediation has long been the focus of scientific interest and is one of the possible therapeutic strategies for the treatment of obesity and appetite regulation [42]. Many different agents that affect the central brain regulation of appetite have been studied over the years. For example, Lorcaserin, a 5-HT2C receptor agonist, was an FDA-approved drug for treating obesity. Lorcase influences the reduction of body weight and blood glucose levels and further cardiovascular benefits [43,44]. However, in 2020, the FDA concluded that the potential risks of lorcaserin outweigh its benefits because of the protentional higher risk of malignancy [45], although there was no clear evidence of that. Due to that, research for new molecules with appetite control effects related to serotonin mediation in the central nervous system continues.

WAY163909 is a selective serotonin receptor agonist (for the 5-HT2C receptor) that we used in our study. In the conducted experiment, a significant reduction in body weight was found in male Wistar rats with induced obesity and type 2 diabetes after intraperitoneal administration of WAY163909, which correlates with the data available so far from the world literature [42]. However, until now, the effect of selective serotonin agonists on blood glucose levels and insulin resistance has not been studied, or data from such studies have been presented rather as an observed effect without being systematically reviewed.

A significant reduction in blood glucose levels was found in obese and diabetic male Wistar rats with a daily intraperitoneal application of WAY163909. These results are comparable with other research using 5-HT2C receptor agonists [42,46]. In the control groups using WAY163909, no hypoglycemic values were recorded, suggesting that the reduction of blood glucose levels is in a glucose-dependent manner. The main question is whether a reduction in blood glucose is due to a reduction in the food intake from the experimental animals or whether there are other mechanisms which can lead to an improvement in carbohydrate metabolisation. This question is ripe for future studies in which obesity is not the main factor for the development of diabetes.

Tracking the levels of immunoreactive insulin and HOMA-index, a significant reduction was also observed in group 1, which was using the selective serotonin receptor agonists and initially had insulin resistance. Although in studies with other 5-HT2C receptor agonists, there was no reduction of immunoreactive insulin levels, yet there was an increased insulin sensitivity, measured with HOMA-index, related to the usage of 5-HT2C receptor agonists [43]. The reduction of the values of the HOMA-index correlates with the reduction of body weight, a phenomenon that was shown in other studies with 5-HT2C receptor agonists [43]. The question is whether this reduction is a consequence of food intake restriction, a phenomenon that was observed in the groups using WAY163909, or whether this reduction in insulin levels has another biochemical and physiological basis that should be investigated.

Peripheral serotonin is also important in the regulation of carbohydrate and lipid metabolism and can affect the function of different organs and cell types. Peripheral serotonin has a direct effect on β-cells in the pancreas by delaying their apoptosis and even stimulating their proliferation while also affecting insulin secretion and blood glucose levels [47,48]. Peripheral serotonin can also have different effects on the liver, directly reflecting carbohydrate and fat metabolism [47,48]. In adipose tissue, serotonin affects lipolysis and glucose uptake and cholesterol ester hydrolysis [47,48] while also controlling thermogenesis by affecting both white and brown adipose tissue [47]. In muscle cells, serotonin mediates glycogenolysis [47]. Due to this, new strategies in the treatment of obesity could be found in a combination of agents that influence both peripheral and CNS serotonin mediation.

Investigation into serotonin mediation through the prism of metabolic disorders is a topic that will be relevant in the coming years [42,48]. An abundance of accumulated evidence concerning the biochemical, physiological, and pathophysiological processes in which serotonin mediation is involved is a main point for the search for new pharmacological agents affecting these processes. In our research, for the first time, we systematically presented the action of brain serotonin mediation in the context of its importance in metabolic processes. Of course, more detailed research is needed on the action of serotonin, both in the peripheral and central nervous systems, to be able to make new classes of medications for the treatment of diabetes, dyslipidemia, obesity, and other metabolic disorders.

## 5. Conclusions

Daily intraperitoneal application of 1 mg/kg of WAY-163909 significantly reduces body weight, hyperglycemia, and peripheral insulin resistance in obese and diabetic Wistar rats. More detailed research is needed on the action of serotonin, both in the peripheral and central nervous systems, to be able to make new classes of medications for the treatment of diabetes, dyslipidemia, obesity, and other metabolic disorders.

## Figures and Tables

**Figure 1 brainsci-13-00545-f001:**
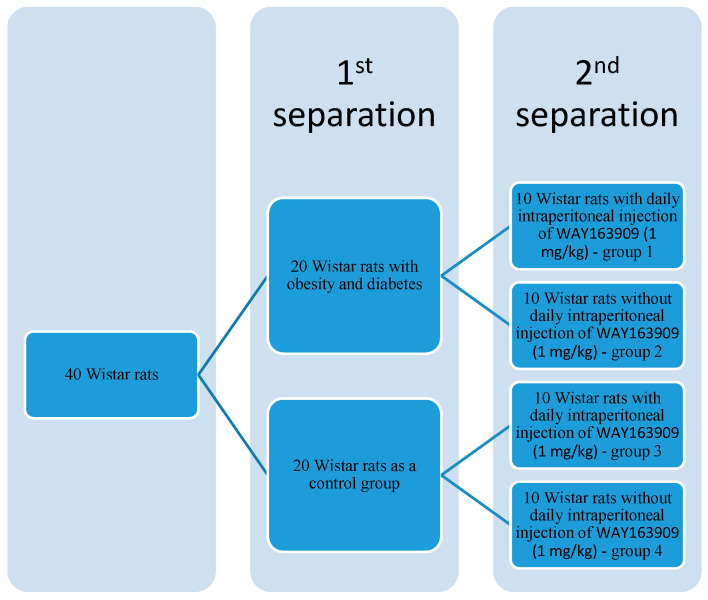
Separation of the groups in the study.

**Figure 2 brainsci-13-00545-f002:**
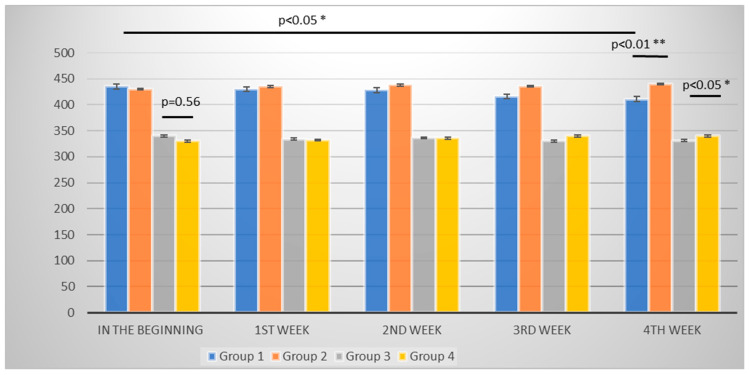
Average body weight, measured in grams, in the different groups, presented by week. Statistical significance: * *p* < 0.05, ** *p* < 0.01.

**Figure 3 brainsci-13-00545-f003:**
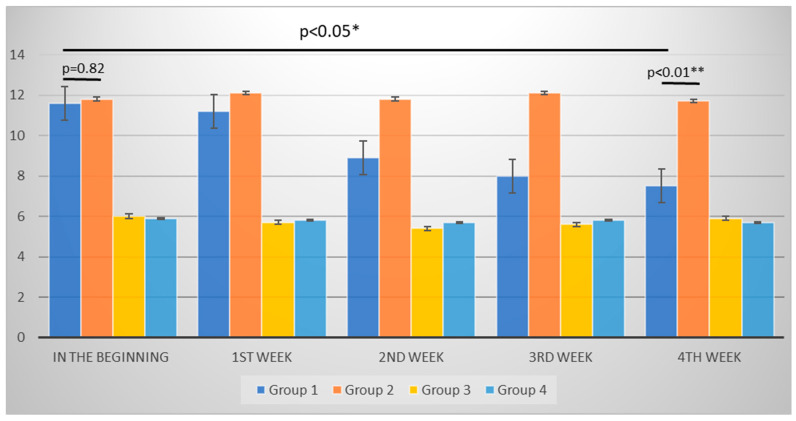
The average levels of blood glucose in the different groups, measured in mmol/L, given by week. Statistical significance: * *p* < 0.05, ** *p* < 0.01.

**Figure 4 brainsci-13-00545-f004:**
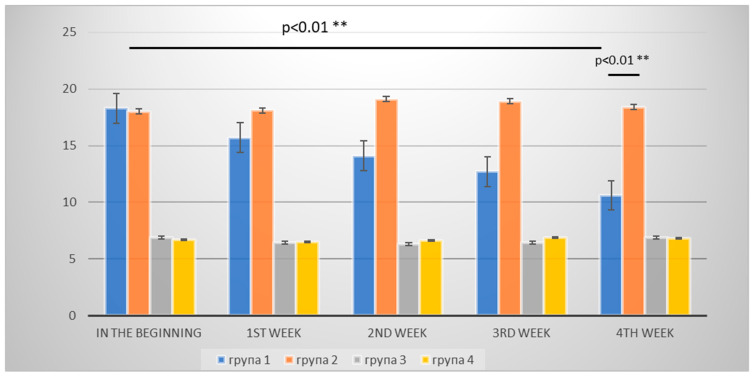
Average levels of immunoreactive insulin, measured in mIU/mL, over the study. Statistical significance: ** *p* < 0.01.

**Figure 5 brainsci-13-00545-f005:**
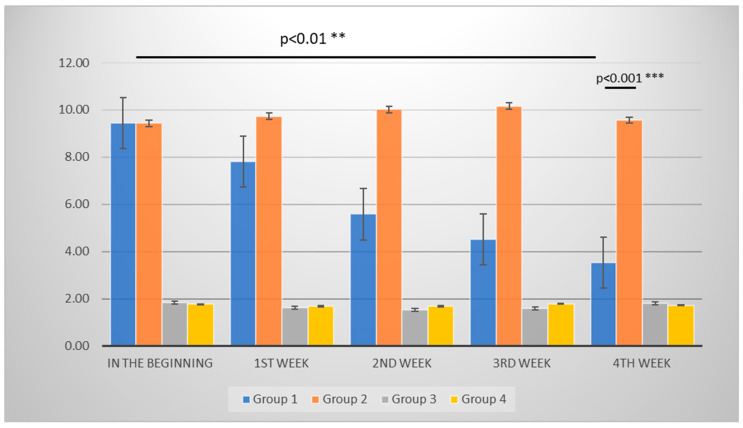
Mean levels of HOMA-index in the different groups during the study. Statistical significance: ** *p* < 0.01, *** *p* < 0.001.

**Table 1 brainsci-13-00545-t001:** Composition and nutritional value of the cafeteria diet.

The Values for 100 g of Food Product	Energy Value, Kcal	Total Amount of Fat, g	Total Amount of Carbohydrates, g	Total Amount of Proteins, g	Sugars, g	Saturated Fatty Acids, g	Salt, g
Chips Lay’s ^®^ Cheese	516	33	48	6.7	2.1	2.6	1.2
7 DAYS Bake Rolls^®^ salt	449	15	63	14	7	5.8	2.7
Crispy soelts with salt Xpyc-Xpyc ^®^	389.5	5.23	77.5	8.11	1.99	1.5	2.8
Crispy baked bread cubes with bacon flavour., Cubeti^®^	456	15	68	9.4	3	1.4	2.2
Crispy waffles, Haя^®^	528.7	28.3	63.1	5.8	41.5	12.6	0.2
Roasted peanuts, Дeтeлинa ^®^	561	41.5	15.5	26.8	4.1	9.1	3
Cereal, NESTLÉ^®^ CHOCAPIC^®^	378	4.4	73.2	9.6	28.9	1.7	0.47

**Table 2 brainsci-13-00545-t002:** The mean body weight values of rats in the different groups during the study ± standard deviation (SD).

	Group 1	Group 2	Group 3	Group 4
in the beginning	435 ± 20.4	430 ± 17.3	340 ± 24.9	330 ± 22.3
1st week	439 ± 20.4	435 ± 17.7	334 ± 23.7	332 ± 21.9
2nd week	428 ± 19.1	438 ± 16.3	326 ± 19.2	336 ± 17.6
3rd week	416 ± 14.1	436 ± 18.3	330 ± 18.2	340 ± 17.1
4th week	411 ± 12.6	441 ± 15.2	331 ± 7.33	340 ± 6.05

**Table 3 brainsci-13-00545-t003:** The average levels of blood glucose levels of the rats in the different groups during the study ± standard deviation (SD).

	Group 1	Group 2	Group 3	Group 4
In the beginning	11.6 ± 2.24	11.8 ± 2.16	6.0 ± 0.46	5.91 ± 0.36
1st week	11.2 ± 1.98	12.1 ± 2.42	5.7 ± 0.47	5.8 ± 0.42
2nd week	8.9 ± 1.49	11.8 ± 1.76	5.4 ± 0.56	5.7 ± 0.61
3rd week	8 ± 1.08	11.7 ± 1.40	5.6 ± 0.80	5.8 ± 0.77
4th week	7.5 ± 0.92	11.9 ± 1.71	5.72 ± 0.42	5.78 ± 0.63

**Table 4 brainsci-13-00545-t004:** Average levels of immunoreactive insulin, measured in mIU/mL, in the different groups during the study ± standard deviation (SD).

	Group 1	Group 2	Group 3	Group 4
In the beginning	18.3 ± 4.1	12.3 ± 3.47	6.9 ± 0.98	6.7 ± 0.86
1st week	15.7 ± 2.92	18.0 ± 3.00	6.4 ± 0.84	6.5 ± 0.89
2nd week	14.1 ± 3.22	19.1 ± 2.74	6.3 ± 0.95	6.6 ± 1.03
3rd week	12.7 ± 2.69	18.9 ± 2.41	6.4 ± 1.19	6.7 ± 0.89
4th week	10.6 ± 1.18	18.4 ± 1.75	6.9 ± 0.77	6.8 ± 1.11

**Table 5 brainsci-13-00545-t005:** Mean levels of HOMA-index in the different groups during the study ± standard deviation (SD).

	Group 1	Group 2	Group 3	Group 4
In the beginning	9.43 ± 1.86	9.44 ± 1.69	1.84 ± 0.33	1.76 ± 0.34
1st week	7.82 ± 1.23	9.73 ± 1.52	1.62 ± 1.62	1.68 ± 0.24
2nd week	5.58 ± 0.54	10.02 ± 1.43	1.51 ± 0.25	1.67 ± 0.24
3rd week	4.58 ± 0.63	10.2 ± 1.07	1.51 ± 0.28	1.67 ± 0.26
4th week	3.53 ± 0.37	9.57 ± 1.02	1.81 ± 0.17	1.72 ± 0.19

## Data Availability

The data that support the findings of this study are available from the corresponding author upon reasonable request.

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
