# Peer review of "New Insight into Selective Serotonin Receptor Agonists in the Central Nervous System, Studied with WAY163909 in Obese and Diabetic Wistar Rats"

_brainsci, 2023, doi:10.3390/brainsci13040545_

Round 1

Reviewer 1 Report

Rewrite sentences 159-160

170 – write the full form if it is used for the first time

170 – Is “ELISA Rat Insulin” Is this the appropriate term to say for the experiment? Use the appropriate terminology for stating the experiment.

Rewrite 172-174…. looks like something is unusual.

2.5.2. Statistical hypothesis testing methods: = please describe this section properly with complete sentences rather than using phrases.

Use past tense in results where necessary

211 = “diabetes type 2”  - write type 2 diabetes and maintain uniformity everywhere.

211, 212 = Did you calculate the initial average body weight between groups 3 and 4? If there was no significant difference in initial body wt then obtaining a significant wt difference at the final would be a good result. So, add a sentence about the initial body wt difference between groups 3 and 4 in 211 and 212.

225,226 rewrite the sentence…hard to get the meaning

Why is the motive of keeping figures 6 and 7 both? Keeping fig 6 only may be sufficient. Please illustrate.

266 -rewrite

Why was the data on food intake not included in the study? Please include it. 

Author Response

Dear sir/madam, 

thank you for your review and excuse me for the long time needed for the answer. I am uploading the new version of the manuscript in which all of your recommendations are taken into account and also data on food intake is added.

Kind regards,

Reviewer 2 Report

The aim of the study was the use of WAY-163909 - a selective serotonin 2C receptor agonist in central nervous system and investigating its effect on body weight, glycemia and insulin resistance in obese and diabetic Wistar rats. Authors to determine it, used the body mass index calculation, biochemical analysis of rat glycemia and determination of insulin resistance by calculation of the HOMA-index.   Abstract -   In lines 14; 15; 16 and 18 it is necessary to adjust the writing of numbers and % without a space and decimal numbers - it should be correct with a period, i.e., 5.5% (p<0.05) and not with comma. Introduction: - in this section, it would be appropriate and, in my opinion, very necessary, to add more recent literary sources from the past 5 years. The used literary sources are older than 10 years. Material and methods section: -   in line 156, it is necessary to add the principle of blood glucose measurement using the glucometer used (this information is missing), it is listed only the type of glucometer. -   similarly, in lines 172-177, it would be appropriate to partially expand the description of the ELISA method so that the method is easily reproducible. In line 177 are missing data such as the type of spectrophotometer and the wavelengths used ... Results -   line 213 and 215 (Figure 2.; Figure 3.) - to display the data from them, it is enough to show only one graph, where will be shown both the average values ​​of body weight and their development during the 4 weeks of the experiment. -   they also lack the names of the graph axes and without indicating the units of the monitored parameter - these need to be supplemented. -   in line 232 and 234 (i.e., Figure 4. and Figure 5.) - adjustments are needed as in the preceding figures 2 and 3 - description of the axes, indication of the units of the measured parameter in the graph and the connection of these two figures into one, where the average values ​​of blood glucose and at the same time, the dynamics of development during the study. Moreover, in the legend, the designation of group 4 is given in the alphabet and not in English. -   the same graph and legend corrections are also needed for Figures 6. (line 257) and Figure 7 (line 258); and also Figures 8. (line 261) and Figure 9. (line 263). -     Discussion - this part of the manuscript will need to be revised; because the authors present here only an evaluation of the issue, but without confronting their results with the findings of other authors. Not a single study or the results of other works focused on the monitored area are cited here - discussion is missing.   Conclusion -   I have no comments. References -   in the manuscript were used 38 literary sources (without self-citations), of which were zero sources from the last 5 years; 1 source is for the last 5-10 years (from yrs. 2014) and 37 sources that are older than 10 years. Literary sources need to be generally updated with newer ones. -   it will also be necessary to modify the references form - line 336 (for literary source 2) and line 362 are empty. -   it would also be appropriate to check and correct the formal page of references in accordance with the journal's recommendations.

Author Response

Dear sir/madam, 

thank you for your review and excuse me for the long time needed for the answer. I am uploading the new version of the manuscript in which all of your recommendations are taken into account.

Abstract - now is used period not a comma as it was a technical mistake. Introduction: - I add recent literary sources from the past 5 years, and in the manuscript, there are more than 50 %. Material and methods section: -  all necessary information was added regarding your recommendation Results - change of the graphics was made as we add more tables to be more clear and the graphics to be more or less better for interpretation   Discussion - we add more information for other studies related with the topic of the manuscripts from the recent years and we tried to make a real discussion in this section.    References -  we update the reference as at least 50% of them are from the last 5 years.

Kind regards,

Round 2

Reviewer 2 Report

Abstract - line 14, 16, 18 - it is necessery to adjuste the writing of numbers  and % without a space

Introduction - in current form it is correct

Material and methods - in current form it is correct

Results - please unify the writing of numerical values ​​with standard deviation in table 2; 3; 4; 5 - table 2 - shows the number and SD without spaces; other tables list these values ​​with spaces

Conclusion - in current form it is correct

References - in current form it is correct

I consider the corrected version suitable for publication in Journal